# Intervention Strategies for Seasonal and Emerging Respiratory Viruses with Drugs and Vaccines Targeting Viral Surface Glycoproteins

**DOI:** 10.3390/v13040625

**Published:** 2021-04-06

**Authors:** Ralph A. Tripp, John Stambas

**Affiliations:** 1Department of Infectious Diseases, College of Veterinary Medicine, University of Georgia, Athens, GA 30605, USA; 2School of Medicine, Geelong Waurn Ponds, Deakin University, Melbourne, VIC 3125, Australia; john.stambas@deakin.edu.au

**Keywords:** respiratory virus, respiratory syncytial virus, RSV, SARS-CoV-2, vaccines, therapeutics

## Abstract

Vaccines and therapeutics targeting viral surface glycoproteins are a major component of disease prevention for respiratory viral diseases. Over the years, vaccines have proven to be the most successful intervention for preventing disease. Technological advances in vaccine platforms that focus on viral surface glycoproteins have provided solutions for current and emerging pathogens like SARS-CoV-2, and our understanding of the structural basis for antibody neutralization is guiding the selection of other vaccine targets for respiratory viruses like RSV. This review discusses the role of viral surface glycoproteins in disease intervention approaches.

## 1. Respiratory Viruses

Acute respiratory tract infections (ARI) constitute a considerable worldwide disease threat and burden that has been made plainly evident by the number of COVID-19 cases caused by SARS-CoV-2 (CoV-2). Usually, influenza virus (flu) is the most widely recognized cause of ARI associated with respiratory illness and substantial disease burden in adults and elderly individuals [1,2]. Beyond CoV-2 and flu there are other respiratory pathogens such as respiratory syncytial virus (RSV), rhinovirus (RV), human metapneumovirus (HMPV), parainfluenza viruses (PIV), and bocaviruses that contribute to worldwide disease burden [2,3]. For example, RSV is associated with considerable disease in older (aged ≥ 65 years) adults and young children [4,5].

The greatest hurdle for control of respiratory viruses is that there are few licensed vaccines and limited antivirals available. Another challenge and concern is that infection with some RNA viruses does not always provide durable protection against reinfection, as is the case with RSV [6], although disease upon reinfection in young adults is often ameliorated and typically restricted to the upper respiratory tract (URT) [7]. This is notable as the goal of immunization is to reduce the severity of disease rather than the induction of sterilizing immunity. It is likely that the induction of long-term protective immunity may require more than one infection and more than one dose of vaccine to augment durable humoral immunity. Viral glycoproteins enable virus entry into the host cell through receptor binding and promote virus egress and the release of progeny. Viral surface glycoproteins are the predominant antigens toward which humoral immune responses are directed. Thus, the principal vaccine strategy is to induce blocking and/or neutralizing antibodies to these immunogens.

## 2. Novel Vaccines

The goal of vaccination is to prevent or reduce disease and control pathogen transmission. Various vaccine approaches may be needed to protect high-risk groups that are at the extremes of age, i.e., very young or old, or have underlying conditions or are immune suppressed in order to provide long-lived acquired immunity. However, as respiratory virus infection chiefly occurs in the respiratory epithelium, serum antibody does not protect the upper respiratory tract [8], and mucosal antibodies wane [9], control of respiratory virus infection is a challenge. Thus, development of efficacious live-attenuated vaccines can be challenging as these vaccines are typically less immunogenic than wild type viruses, and if wild type viruses do not induce durable immunity, it is unrealistic to expect attenuated vaccines to improve the vaccination outcome. This is relevant because attenuation and immunogenicity are linked, and these features narrow the therapeutic window. Based on clinical vaccine trials for viruses like RSV, multiple doses of vaccine seem to be needed to achieve practical immunity.

RSV is a negative-sense, non-segmented enveloped RNA virus causing substantial morbidity and some mortality in adults and children [10,11]. RSV contributes to an estimated 3 million hospitalizations and >60,000 deaths/year globally [12]. Unfortunately, no safe and approved RSV vaccine exists despite numerous attempts using various vaccine strategies in recent decades. The most infamous vaccine strategy for RSV was the formalin-inactivated RSV vaccine (FI-RSV), which induced enhanced disease in young vaccinees when they were subsequently naturally infected with RSV [13,14]. The FI-RSV vaccine caused unexpected enhanced respiratory disease (ERD) and some mortality. Later it was suggested that this poor vaccine outcome was likely linked to the failure to induce neutralizing antibodies, and the failure to generate efficient IFN and CTL responses [15,16]. The outcome of the FI-RSV vaccine trials raised regulatory and safety concerns ultimately hampering RSV vaccine development.

Important facets that affect RSV vaccine development are the absence of an ideal animal model and the lack of established correlates of protection. The RSV vaccine conundrum is complicated as no animal model fully replicates RSV disease, and all older adults are seropositive but not necessarily protected. The field has relied on preclinical models, particularly BALB/c mice, to develop correlates of protection that make it difficult to define immunogenicity endpoints as RSV is only semi-permissive in this model. Recently, there has been a substantial effort to develop RSV vaccines that incorporate the fusion (F) protein that facilitates viral fusion with host cells [17,18]. The other major RSV surface glycoprotein, i.e., the attachment (G) protein, is also being explored but is thought to be a less promising vaccine candidate due to gene variability between strains [18,19]. The main differences between RSV A and RSV B strains is the amino acid sequence of the G protein. The G protein is a type II membrane protein having an extracellular ectodomain containing four cysteine residues that are highly conserved in all RSV isolates. This central conserved region (CCR) contains a CX3C chemokine motif (amino acids 182–186) that promotes RSV attachment to susceptible cells bearing a CX3C chemokine receptor, CX3CR1 [20]. Thus, G protein CCR-based vaccine candidates remain of interest.

The F protein is conserved between RSV strains, is immunogenic, and provides protection against RSV challenge in preclinical studies [21]. The F protein has two conformational states, i.e., pre-fusion and post-fusion forms [22]. The F proteins of A and B strains are approximately 90% identical in amino acid sequence [23]. However, the pre-F and post-F proteins have shortcomings as antibodies elicited to post-F have been shown to neutralize fusion activity, however the post-F protein lacks key epitopes found in pre-F protein, including antigenic site Ø which changes during transition from pre-F to post-F [24]. Accumulating evidence suggests that neutralizing antibodies against the F and G proteins of both RSV subtypes could induce protection against RSV, as it has been shown that the addition of the G protein to F protein-based vaccines enhances RSV protective antibody titers [25,26].

Influenza viruses are negative sense, single-stranded enveloped RNA viruses with segmented genomes that are responsible for approximately 250,000–500,000 deaths annually [27]. Current prevention strategies include annual vaccination based on circulating influenza virus strains and use of antiviral drugs. Influenza A viruses express 13 proteins from 8 gene segments (HA, NA, NP, PB1, PB2, PA, NS, and M) three of which are found on the surface, i.e., the hemagglutinin (HA) and neuraminidase (NA) glycoproteins, and the integral membrane M2 ion channel protein. The HA and NA glycoproteins are used to subtype influenza A viruses. There are currently 18 HA and 11 NA subtypes. Among the >130 influenza subtype combinations, only the H1N1 and H3N2 subtypes circulate in humans. HA is expressed on the virus surface as a trimer and binds sialic acid residues on host cells to facilitate virus entry. After binding, the virus is internalized via an endosome. The HA then undergoes a conformational change upon acidification of the endosome and fuses with the endosomal membrane to release the virus genome into the cytoplasm of the host cell to begin the replication cycle. NA is expressed as a tetramer on the surface of the virus and has an active site situated in the globular head that is used to release influenza progeny following completion of the replication cycle [28]. Both the HA and NA serve as key targets for antibodies following infection or vaccination.

There are currently two types of influenza vaccines commercially available for use in humans: (a) inactivated and (b) live-attenuated vaccines, both of which stimulate protective antibody responses [29]. Inactivated vaccines are administered intramuscularly and are produced in trivalent or quadrivalent forms that have been prepared by chemical inactivation or detergent disruption to stimulate antibody responses [29]. Live-attenuated vaccines mimic natural infection but are cold-adapted and replicate only in the upper respiratory tract to produce both humoral and cellular immune responses [30]. This platform is thought to encourage cross-protection against heterologous strains of virus. Recently, universal vaccine platforms have focused on the conserved HA stem region to produce HA-stem specific antibodies that have the potential to be cross-reactive [31]. This is in contrast to antibodies generated against the highly variable HA globular head (current influenza vaccines) which requires annual reformulation of vaccines to ensure vaccine efficacy.

Vaccine development is a lengthy process that is intolerable during a pandemic such as that caused by SARS-CoV-2. Briefly, the process begins with vaccine design and evaluation in animal models, it is then followed by preclinical experiments and toxicology studies, before an application is filed with the FDA for use as an investigational new drug. Thereafter, Phase I, II, and III clinical trials are established to confirm immunogenicity and dose, and evaluate safety and efficacy. If the outcome of Phase III trials meets pre-defined endpoints, a biologics license application is filed with the FDA. The SARS-CoV-2 pandemic required rapid development of vaccines in an unprecedented timeframe (Operation Warp Speed, [32,33,34]. As a result, >200 vaccine candidates based on several different platforms are currently in various stages of development [35,36], and 3 RNA vaccines have been approved and are in use worldwide. RNA vaccines that express the spike (S) protein of SARS-CoV-2 delivered via lipid nanoparticles have shown great promise [37,38]. The S protein is the coronavirus surface protein that mediates entry into susceptible cells. The S protein binds to its receptor, angiotensin-2 (ACE2) through its receptor-binding domain (RBD) and is proteolytically-activated by human proteases [39]. Thus, inhibiting S protein binding to ACE2 by antibodies is the current goal of SARS-CoV-2 vaccines. The current SARS-CoV-2 vaccine candidates have produced a wide spectrum of immunogenicity and neutralizing antibodies.

## 3. Novel Drugs

Therapeutic intervention of respiratory virus infection and replication has involved strategies that target virus entry mediated by viral surface glycoproteins. There are several antiviral drugs that act as virus entry inhibitors, protease and transcriptase inhibitors, maturation and virus particle inhibitors, all of which affect critical steps of the virus life cycle. Notably, blocking of viral surface glycoproteins can prevent host cell receptor attachment, virus internalization, and infection [40]. Some RNA viruses have an envelope made of virus-modified host membrane. These viruses are grouped into families based on several criteria, including how they replicate, their assembly, and the structure of their virions. For replication, viruses attach to the plasma membrane receptors via their cell surface envelope proteins. This process involves initial binding leading to conformational changes in the fusion protein, insertion into the plasma membrane, and release of the virus genome into the cytoplasm. Enveloped viruses have non-specific interactions with host cell adhesion molecules, e.g., glycosaminoglycans, sialic acid, etc. until they encounter their specific entry receptors. Fusion with the host cell membrane then allows the virus to release its genome into the host cell cytoplasm. Viruses create syncytia that involve the host plasma membrane to facilitate cell entry a process leading to conformational changes in their envelope protein [41]. Very few drugs have been approved for human use that function as envelope protein inhibitors [42]. Many drug therapies are currently exploring the use of small molecule drugs [18], however treatment for RSV is currently limited to supportive care and prophylactic antibody treatment using the prophylactic neutralizing antibody, palivizumab [4]. Palivizumab is a humanized IgG mAb that targets pre-fusion RSV F protein [43]. Palivizumab therapy is recommended for premature infants or for infants <2 years old with chronic lung diseases or heart disease. The use of Palivizumab is limited to at-risk groups.

Influenza virus HA is a favorable target for anti-influenza drugs but effective HA inhibitors have not been developed due to the emergence of drug-resistant viruses with amino acid substitutions on the HA [44]. Although antivirals are not commercially available to target the HA, several antivirals are capable of inhibiting NA activity. Influenza virus progeny are normally released from infected cells by the enzymic activity of NA which cleaves sialic acids on cell surface receptors [45]. Two antiviral neuraminidase inhibitors, oseltamivir and zanamivir, were first discovered in the 1990s [46,47] and licensed in 1999 for human use. Both drugs were rationally designed based on the crystal structure of NA, and from studies involving DANA, an inhibitor of neuraminidase activity [47,48]. Both zanamivir and oseltamivir are highly effective against influenza A and B viruses. Zanamivir had poor bioavailability when delivered by the oral route and as such was developed as a powder that is administered by inhalation [49]. The design of oseltamivir was based on zanamivir but incorporated changes that improved oral bioavailability [46]. Another neuraminidase inhibitor, Peramivir, has structural features that distinguish it from zanamivir and oseltamivir [50]. However due to poor oral bioavailability, intravenous delivery of Peramivir was shown to found to be effective [51].

It is important to note that other antivirals have been shown to be effective in targeting the influenza life cycle are also approved and available for use by clinicians. The matrix 2 protein inhibitors, amantadine and rimantadine were first shown to be effective against influenza A [52] but have no impact on influenza B virus infection [53]. This class of antivirals act by interrupting M2 ion channel function and prevent uncoating of influenza A viruses and release of ribonucleoproteins [53]. Due to high rates of resistance, their clinical use is very limited. Recently, the endonuclease inhibitor baloxavir has also been approved for human use. Baloxavir is taken orally and binds to the cap-dependent endonuclease that results in disruption of viral mRNA synthesis and influenza virus replication [54]. Thus, several antiviral drugs can ameliorate influenza virus symptoms following infection including oseltamivir (orally), zanamivir (inhalation), peramivir (intravenously) and baloxavir (orally).

Antiviral drug use at early timepoints after influenza virus exposure is appropriate for most individuals, however widespread prophylactic administration is only recommended for specific settings e.g., aged care where established outbreaks place vulnerable patients at risk and where administration of the antivirals is being used as part of a larger containment strategy [55]. Surveillance of circulating viruses is needed to ensure resistance to antivirals does not occur. For example, amantadine resistance to H3N2 influenza viruses (S31N dominant mutation) led to withdrawal of recommendations for clinical use [56]. Oseltamivir resistance has been noted in H1N1 influenza A viruses (H275Y dominant mutation) although its frequency is relatively low in current circulating strains [57]. In 1999, antiviral resistance concerns led to the establishment of the global Neuraminidase Inhibitor Susceptibility Network (NISN) to monitor susceptibility and resistance from global isolates [58]. Combination antiviral therapies are now being considered as a way of limiting the emergence of antiviral resistance with baloxavir and neuraminidase inhibitor combinations showing promise in a human airway epithelium model [59]. Moreover, studies in mice show that combination therapies involving baloxavir and the neuraminidase inhibitor oseltamivir can reduce mortality associated with H1N1 PR8 infection when compared to single therapy alone [60].

As the S protein of SARS-CoV-2 has a key role in the receptor recognition and cell membrane fusion process, it is being targeted by antiviral drugs [61,62]. Several SARS-CoV-2 entry inhibitors such as monoclonal antibodies, recombinant peptides and proteins, and small molecule antivirals, as well as repurposed drugs are being evaluated. The S protein is composed of two subunits, S1 and S2. The S1 subunit contains a receptor-binding domain that recognizes and binds the host receptor angiotensin-converting enzyme 2, while the S2 subunit mediates viral cell membrane fusion by forming a six-helical bundle via the two-heptad repeat domain. Most therapeutic monoclonal antibodies bind to S1 and block its binding to ACE2 on target cells [63]. In addition, convalescent plasma contains neutralizing antibodies that bind to different regions on S protein and inhibit viral entry by different mechanisms [64,65]. Moreover, various synthetic molecules bind to S protein and inhibit binding of to ACE2, block SARS-CoV-2 fusion with the cell membrane [66], or inhibit cleavage of polyproteins required for SARS-CoV-2 replication such as viral proteases, 3CLpro and PLpro [67]. Remdesivir and favipiravir are two of the most promising antiviral drugs that have been tested [68,69]. Numerous clinical trials are currently ongoing worldwide to evaluate antiviral drug efficacy and disease management in COVID-19 patients. All of these trials aim to decrease morbidity and mortality until a specific drug is developed.

## Data Availability

Not applicable.

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
