# Peer review of "Intervention Strategies for Seasonal and Emerging Respiratory Viruses with Drugs and Vaccines Targeting Viral Surface Glycoproteins"

_viruses, 2021, doi:10.3390/v13040625_

Round 1

Reviewer 1 Report

The revie manuscript by Tripp and Stambas discusses vaccination and therapeutic strategies targeted to the surface glycoproteins of respiratory viruses. The manuscript is overall well written and covers three important viruses. I only have some minor comments:

  • The authors claim that primary respiratory infection does not induce robust immunity/protection from re-infection (lines 32-34 & 50-51). I think these statements need to be revisited and supported by appropriate citations. There are plenty of studies demonstrating that infection does induce long-lived immunity that can be associated with protection, re-infection is more often a result of antigenic drift - as well as waning immunity to some extent but this does not equal the lack of robust immunity during infection.
  • Line 55 “ vaccines can is challenging” has an error

Author Response

Reviewer 1:

  • We have modified text to highlight that for some RNA viruses (e.g. RSV) immunity is incomplete and re-infection is common (PMID: 10885982). We have added references to support this.
  • We have corrected the typo as requested

Reviewer 2 Report

This review covers the use of viral spike glycoproteins either as candidate vaccine antigens and also as drug targets. The review is succinct, useful and of course timely.

One area that should be addressed is the lack of referencing throughout. For example there is a paragraph discussing influenza vaccines with no absolutely no referencing of the primary literature (lines 117-128)

Author Response

Reviewer 2:

  • We have added references to the section indicated as requested. There are 69 references in all in this mini-review.